# A Comprehensive Evaluation Model of Ammonia Pollution Trends in a Groundwater Source Area along a River in Residential Areas

Zhuoran Wang [1,*], Xiaoguang Zhao [1,2], Tianyu Xie [3], Na Wen [1] and Jing Yao [1]

1 College of Geology and Environment, Xi'an University of Science & Technology, No. 58 Yanta Road, Xi'an 710054, China; sxbxylz@163.com (X.Z.); wrp1024@126.com (N.W.); jingling89757@yahoo.com (J.Y.)

2 Institute of Mine Environmental Engineering, Xi'an University of Science & Technology, No. 58 Yanta Road, Xi'an 710054, China

3 Guodu Education Science and Technology Industrial Development Zone, School of Art, Xi'an International Studies University, No. 6 Wenyuan South Road, Xi'an 710128, China; Xie_Tianyu@126.com

* Correspondence: 19109071013@stu.xust.edu.cn; Tel.: +86-187-9140-8020

**Abstract:** In this study, a comprehensive evaluation model of ammonia pollution trends in a groundwater source area along a river in residential areas is proposed. It consists of coupling models and their interrelated models, including (i) MODFLOW and (ii) MT3DMS. The study area is laid in a plain along a river, where a few workshops operate and groundwater is heavily contaminated by domestic pollutants, agricultural pollutants, and cultivation pollutants. According to the hydrogeological conditions of the study area and the emissions of ammonia calculated in the First National Pollution Source Census Report in China, this study calibrates and verifies the prediction model. The difference between the observed water level and the calculated water level of the model is within the confidence interval of the test. This means that the model is reliable and that it can truly reflect changes in the groundwater flow field and can be directly used to simulate the migration of ammonia. The simulation results show that, after 20 years, the center of the ammonia pollution plume will gradually flow east along with the groundwater over time, mainly affecting the groundwater, which is less than 200 m from the river, and the ammonia content near wells at a maximum extent of less than 0.3 mg/L.

**Keywords:** ammonia pollution; model; MODFLOW; pollution plume; groundwater

## 1. Introduction

Due to rapid growth of the global population, in order to meet growing food demands, the use of chemical fertilizers is also increasing, with nitrogen fertilizer consumption reaching almost 60 million t and occupying nearly 1/3 of the world total consumption in 2016, which is 5 times and 2.5 times than that in the years 1985 and 2005, respectively [1–3]. In some areas, nitrogen-containing pollutants are discharged into rivers. In this water environment, the main types of nitrogen are ammonia, nitrite nitrogen, and nitrate nitrogen [4–6]. For example, after 20 years of Rhine River governance, although the ammonia concentration in the river water decreased by 76%, the nitrate concentration increased by approximately 35% because of the extensive use of chemical fertilizers [7]. Ammonia is toxic to aquatic organisms; it consumes oxygen in water and destroys the balance of the original ecosystem [8,9]. High concentrations of ammonia can affect human eyes, throat, respiratory tract, etc. and can cause bronchitis, pneumonia, pulmonary edema, even coma, and shock, posing non-carcinogenic risks to the human body [10].

Obviously, groundwater resources are one of the main sources of water for industrial and agricultural purposes and in daily life [11–13]. Especially in some arid and semi-arid areas, groundwater resources are the only source of water supply [14–17]. For example, in China, domestic water consumption of 33% of the cities in arid areas are totally dependent on groundwater resources. Therefore, a large number of researchers have studied the

migration of pollutants between rivers and groundwater. Petry demonstrated that the nitrate concentration is mainly controlled by hydrological conditions [18]. Ohte and Martin showed that the groundwater nitrate concentration distribution controls seasonal nitrate variation in the stream [19]. Lapworth showed that shallow groundwater is both a source and a sink for dissolved N and that deoxidation conditions of riparian areas are important in N transformation [20].

When river water replenishes groundwater, the solute in river water enters groundwater or coastal pumping wells through a river infiltration system by nature. In this process, pollutants in the river water are affected by convection–diffusion, adsorption–desorption, biodegradation, plant absorption, etc., and their concentration is reduced or eliminated; thus, the direct impact of a river pollutant on groundwater quality is lessened. The functions related to this process are shown in Figure 1. There are much research on sewage purification by a river filtration system. For example, Jürgen Schubert studied the filtration system of Rhine River and found that the removal rates of ammonia and nitrite nitrogen through nitrification and denitrification are very high [21]. Klaus-Dieter Balke et al. found that pollutants are reduced by natural attenuation in the process of river water infiltration and that clay minerals, iron hydroxide, humus, and underground microorganisms have better decontamination abilities [22].

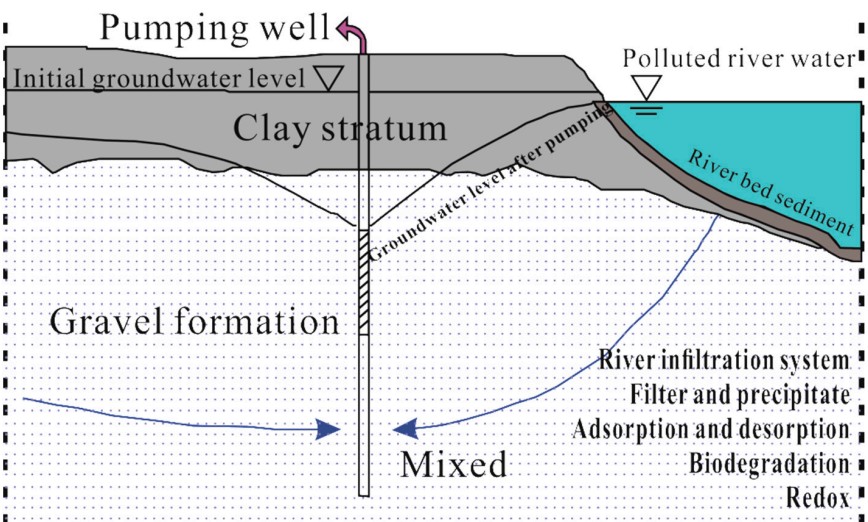

**Figure 1.** Schematic diagram of the migration and transformation of pollutants in a river infiltration system.

Alvarez-Cobelas et al. studied nitrogen (N) export rates from 946 rivers around the world as a function of quantitative and qualitative environmental factors such as land-use, population density, and dominant hydrological processes. They concluded that regional modeling approaches are more useful than global large-scale analyses [23]. Coupled models have thus been developed and used since the 1980s to simulate N transformation at the field scale (SOILN [24], WAVE [25], LEACHN [26], and CREAMS [27]) or nitrate transfer at the catchment scale. Haizhu Hu et al. established a two-dimensional finite element steady flow numerical simulation with FEFLOW and simulated the migration and transformation law of $NO^{3-}$, $NH^{4+}$, DOC, and DO solutes in a river–groundwater system under the condition when phreatic water supplies river water. Hashem A. Faidi pointed out that the simulation of river–groundwater flow usually includes two parts: flow simulation and solute transfer, and water exchange or solute exchange between the river and groundwater system.

Thus, taking the Feng River water source as the research object, this study simulates and analyzes the hydrological and water quality characteristics and the trends of groundwater ammonia pollutant migration in the study area using MODFLOW and MT3DMS in

GMS (Groundwater Modeling System). Using field research and consulting the documents and of hydrogeological condition in the area, a mathematical model of hydrology and water quality is established and used in GMS to correct and validate the main parameters. This particular study is beneficial to groundwater protection and provides a basis for scientific development and utilization [28–32].

## 2. Materials and Methods

### 2.1. Study Area

The Feng River, a water source located in the west suburb of Xi'an, Shaanxi Province, in China, is at 108°35′–109°09′ east longitude and 33°50′–34°20′ north latitude (Figure 2). It boasts a basin area of 1380 km², an average altitude of about 380 m, and an average annual precipitation of 621 mm. The study area of the model is 41.94 km². The groundwater source area mainly supplies phreatic water, which is the only kind of water considered in our simulation. The groundwater in this area is pore water formed by the quaternary system and can be simulated approximately as pore phreatic water.

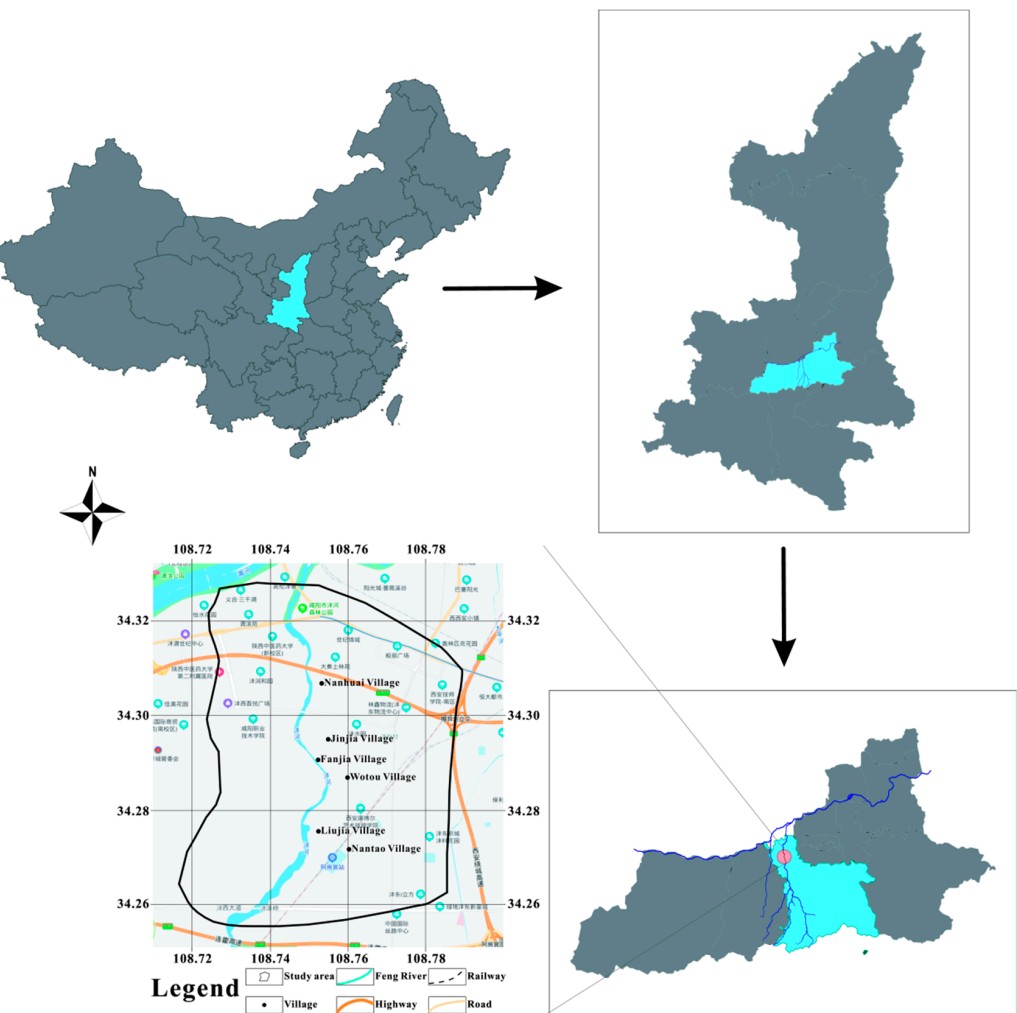

**Figure 2.** Study area.

### 2.2. Numerical Simulation of Groundwater Flow

2.2.1. Generalization of Aquifers and Boundary Conditions

The sand layer under the loess on the surface of the simulation area is generalized as the phreatic aquifer; the lower silty clay layer is a great impervious bed as its water permeability is weak. The vertical movement of the water flow is not considered, and

the water flow is generalized as the two-dimensional plane flow. Different lithological characters of rock mass in the aquifer lead to different hydrogeological parameters of the aquifer, but with basically the same flow directions, the aquifer is generalized as the heterogeneous isotropic medium. Since the silty clay layer is a great impervious bed due to its weak water permeability, it is generalized as the zero-flux boundary. The major drinking water in the study area is the shallow groundwater with small buried depth, small hydraulic gradient, and weak permeability. The groundwater here is mainly replenished by river water and rain and is discharged by outflow and exploitation. As a function of time, the groundwater flow varies as rainfall duration changes, so it is generalized as the unsteady flow [33].

The flow direction of the groundwater is roughly the same as that of the river, and both of them run from southwest to northeast. On the east and west sides of the simulation area, the vertical line of the contour of the groundwater is viewed as the boundary impervious to water, while the north and south sides of the simulation area are defined as the inflow boundary and the outflow boundary, respectively (Figure 3).

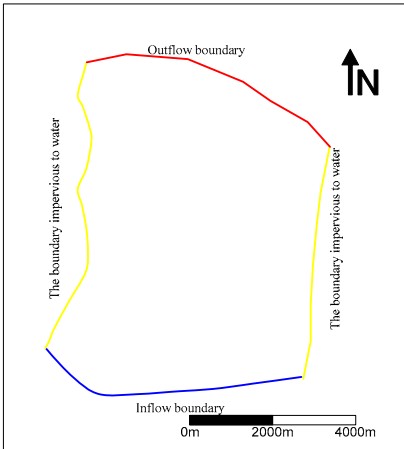

**Figure 3.** Generalization of the boundary conditions.

### 2.2.2. Meshing

The meshing distances of the model grid in the X, Y, and Z directions are 6850 m, 8060 m, and 40 m, respectively. The meshing cell is 50 m × 50 m. The model boundary is imported into the Groundwater Modeling System (GMS), displaying dark gray and light gray. The dark gray represents active cells, and the light gray represents inactive cells (Figure 4).

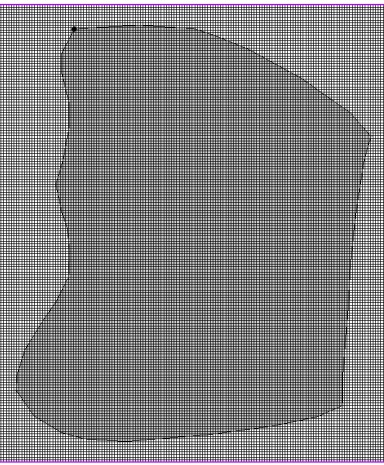

**Figure 4.** Meshing.

2.2.3. Terms of Source and Sink

After passing through the unsaturated zone, rain can directly enter the phreatic water. The formula of the simulated rain recharge is as follows:

$$Q_{rain} = \sum_i \alpha_i P_i A_i \tag{1}$$

where each part of the formula is as follows:

$Q_{rain}$—quantity of rain recharges (m$^3$/d);
$\alpha_i$—coefficient of rain infiltration of each calculation partition;
$P_i$—precipitation of each calculation partition (m/d); and
$A_i$—area of each calculation partition (m$^2$).

Due to the heterogeneity of the precipitation infiltration supply caused by different hydrogeological conditions in the study area, infiltration zoning should be used to generalize the precipitation infiltration supply conditions. According to the data of precipitation infiltration in the study area, the depth of the buried water level, and the lithology of the vadose zone, the zoning figure of precipitation infiltration in the study area can be drawn (Figure 5).

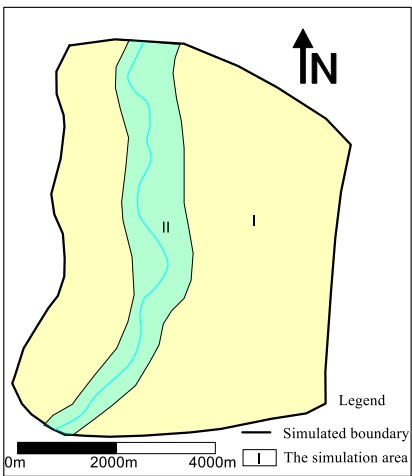

**Figure 5.** Partition of the coefficient of rain replenishment.

Combined with the buried depth of the groundwater level and the lithology of the unsaturated zone, the precipitation infiltration coefficient in the simulation study area was finally determined as shown in Table 1.

**Table 1.** Values of the coefficient of rain replenishment in the simulation area.

| Calculation Partition | Parameter Value | Calculation Partition | Parameter Value |
|:---:|:---:|:---:|:---:|
| I | 0.10 | II | 0.08 |

The data show that the level of groundwater is lower than that of the river, so the groundwater is replenished by the river. The quantity of replenishment is as follows:

$$Q_{river} = K \cdot B \frac{H^2 - h^2}{2b} \tag{2}$$

where each part of the formula is as follows:

$Q_{river}$—quantity of river recharges (m$^3$/d);
K—coefficient of infiltration (m/d);
B—length of the river in the study area (m);
H—distance from the river level to the aquifer floor in the calculation interval (m);

h—distance from the phreatic water level to the aquifer floor in the calculation interval (m); and

b—width of the supply zone of the river (m).

The quantity of lateral runoff recharge and discharge is as follows:

$$Q_{Lateral} = \sum_i K_i I_i A_i \tag{3}$$

where each part of the formula is as follows:

$Q_{Lateral}$—quantity of lateral runoff recharges (m$^3$/d);

$K_i$—coefficient of infiltration of the aquifer of part i;

$I_i$—normal hydraulic slope of the section of part i; and

$A_i$—area of the section of the aquifer of part i (m$^2$).

The irrigation return flow recharge is as follows:

$$Q_{irrigation} = Q_{agri}\beta \tag{4}$$

where each part of the formula is as follows:

$Q_{irrigation}$—quantity of well irrigation return flow (104 m$^3$/a);

$Q_{agri}$—quantity of agricultural extraction (104 m$^3$/a); and

$\beta$—coefficient of well irrigation returns flow.

According to the data provided by waterworks, there are 26 total wells in the groundwater source area, 21 of which are in use, with a produced quantity of 20,000 m$^3$/d. According to Aviriyanover, the maximum depth of evaporation from the buried phreatic water varies and ranges from 1.5 m to 4.0 m. As the depths of the buried phreatic water in the study area are all over 4 m, the evaporation capacity is viewed as zero (Figure 6).

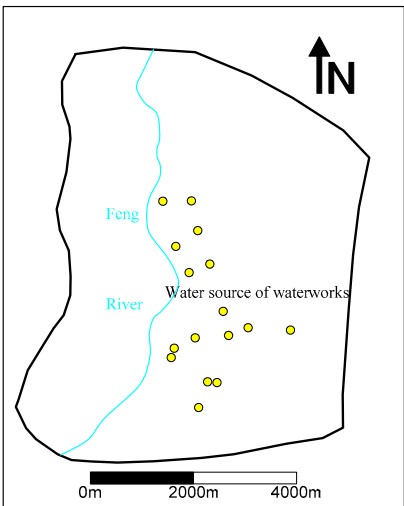

**Figure 6.** Positions of the river and wells.

2.2.4. Hydrogeological Parameter

According to the data from the pumping test provided by waterworks, the average permeability coefficient along the Feng River is 29.14 m/d. The coefficient to the north of the Feng River with bigger peddles is 49.16 m/d and that in the south of the Feng River with smaller pebbles is 10.32 m/d.

The leakage coefficient is in the range between $3.6 \times 10^{-7}$ and $2.5 \times 10^{-6}$ (1/d). The simulation partitions (Figure 7) and the initial parameters of each partition are displayed in Table 2.

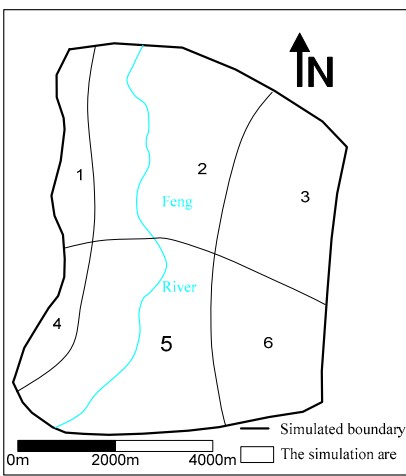

**Figure 7.** Parameter partitions.

**Table 2.** Initial parameter values of the model.

| Number | Horizontal Permeability Coefficient (m/d) | Specific Yield | Number | Horizontal Permeability Coefficient ((m/d) | Specific Yield |
|---|---|---|---|---|---|
| 1 | 45.2 | 0.2 | 4 | 6.4 | 0.2 |
| 2 | 38.3 | 0.2 | 5 | 16.1 | 0.3 |
| 3 | 35.2 | 0.2 | 6 | 7.8 | 0.3 |

The radius of influence to the north of the river is viewed as 237.5 m, that in the south of the river is seen as 52.3 m, and the specific yield (μ) of the aquifer is regarded as between 0.2 and 0.3.

### 2.2.5. Mathematical Model

$$
\begin{cases}
\dfrac{\partial}{\partial x}\left\{K[H - Z(x,y)]\dfrac{\partial H}{\partial x}\right\} + \dfrac{\partial}{\partial y}\left\{K[H - Z(x,y)]\dfrac{\partial H}{\partial y}\right\} + \varepsilon = \mu\dfrac{\partial H}{\partial t} & (x,y) \in \Omega,\ t > 0; \\[2mm]
H(x,y,t)|_{t=0} = H_0(x,y) & (x,y) \in \Omega,\ t > 0; \\[2mm]
K_n\dfrac{\partial H}{\partial n}\bigg|_{\Gamma_2} = q(x,y) & (x,y) \in \Gamma_2,\ t > 0.
\end{cases}
\tag{5}
$$

where each part of the model is as follows:

$\Omega$—vadose zone;

H—height of the groundwater level (m);

K—hydraulic conductivity of the aquifer in horizontal direction (m/d);

$H_0$—initial flow field (m);

$\varepsilon$—terms of source and sink of the aquifer (m/d);

$\Gamma_2$—second-class boundary of the vadose zone;

n—normal direction of the boundary surface;

$\frac{\partial H}{\partial n}$—derivative of H along the outer n (dimensionless);

q—single-width flow on $\Gamma_2$ ($m^2$/d), with its inflow positive and its outflow negative; and

Z(x,y)—elevation of the aquifer floor (m).

### 2.2.6. Solving Process

The MODFLOW solving process is shown in Figure 8.

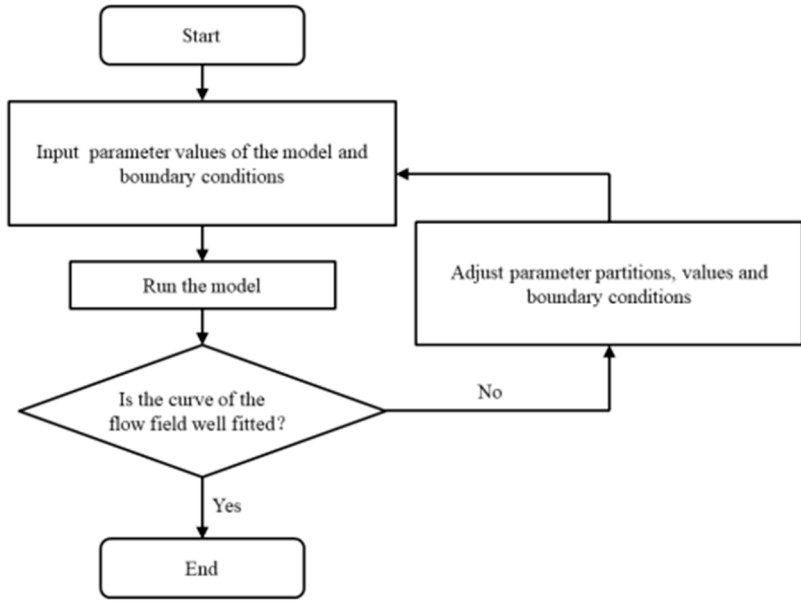

**Figure 8.** MODFLOW solving process.

### 2.2.7. Temporal Discretization

The period from 1 January 2020 to 31 December 2020 was selected to verify the model. Each corresponding calendar month was taken as a stress period, and then, the whole simulation period was divided into 12 stress periods. The step length of each stress period was 10 days. After the model was identified and verified, the corresponding information was input and output. Then, the dynamic changes in the groundwater level in the next 20 years from 1 January 2021 to 31 December 2040 in the groundwater source area were simulated and predicted.

### 2.2.8. Run the Model

Under the condition of normal pumping in the groundwater source area, Figure 9 stimulates the contours of water for different periods of 3 years, 5 years, 10 years, 13 years, 15 years, and 20 years. The overall changes in the groundwater flow field in the next 20 years can be seen. Influenced by precipitation, evaporation, the amount of pumping, and local hydrogeological conditions, the groundwater level in the study area changes obviously. Additionally, the long-term exploitation of groundwater will lead to the cone of depression of the groundwater level around the exploitation site.

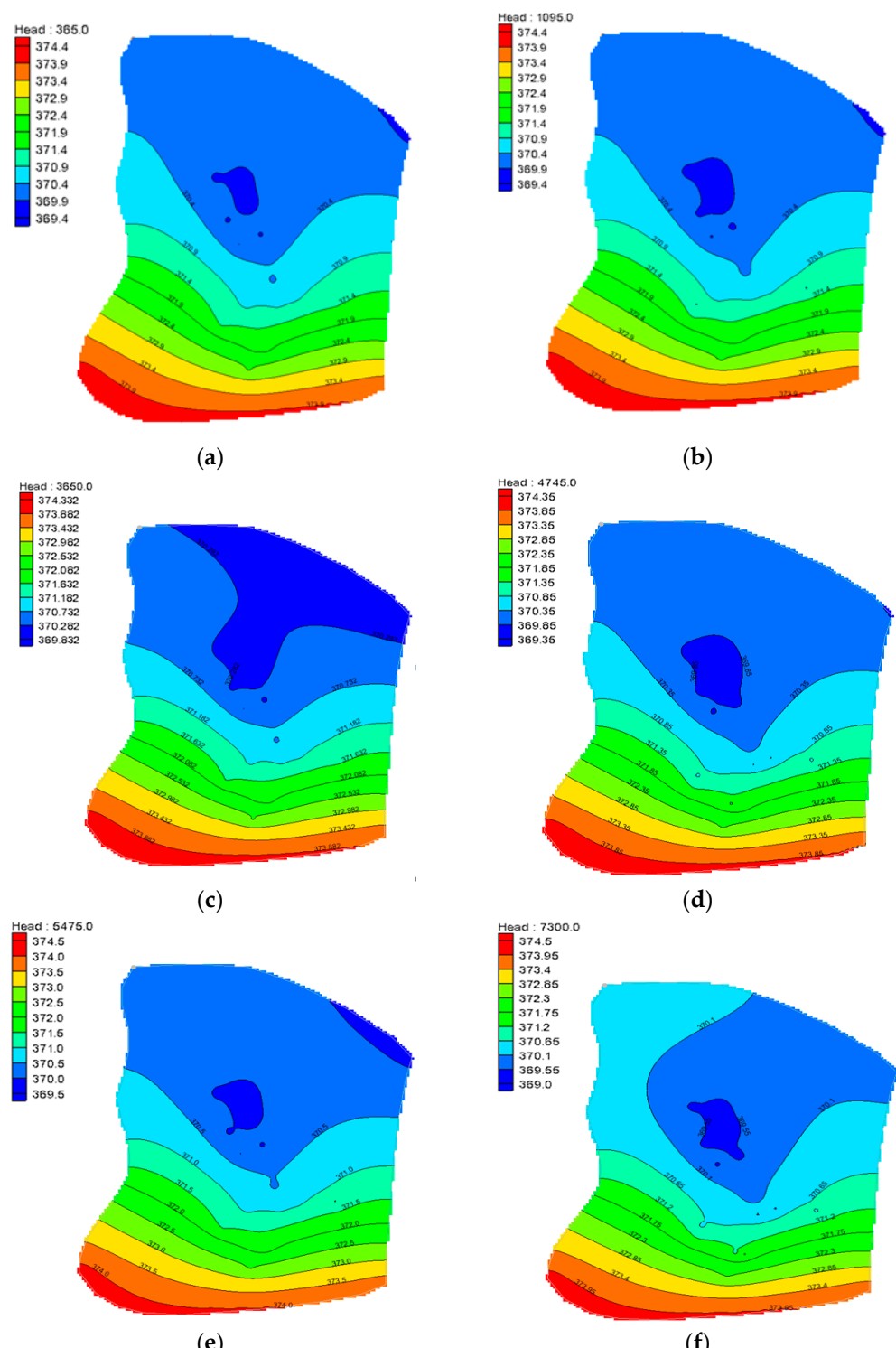

**Figure 9.** Distribution of water level and flow field at different simulation times. (**a**) Simulates the water level after 3 years under nomal pumping conditions (**b**) Simulates the water level after 5 years under nomal pumping conditions (**c**) Simulates the water level after 10 years under nomal pumping conditions (**d**) Simulates the water level after 13 years under nomal pumping conditions (**e**) Simulates the water level after 15 years under nomal pumping conditions (**f**) Simulates the water level after 20 years under nomal pumping conditions.

### 2.2.9. Model Identification and Validation

The simulation controls the fitting precision by virtue of controlling and observing the dynamic changes in the groundwater level and collects data about dynamic changes in the groundwater level from 2019 to 2020 to identify the model. There are six wells for water level observation that can offer data, and all of them are located on the east bank of the Feng River. According to the position of the error bars of the data and the colors, which show how the model is adjusted, when the error bars are above the observed value, they indicate that the actual water level is lower than the calculated water level while, when they are below the observed value, they show that the actual water level is higher than the calculated water level. When the data for the calculated water level and the actual water level are within the confidence interval, the error bars are filled with green. When the data are above the confidence interval but by no more than 200%, the error bars are filled with orange. Additionally, when it is above 200%, the error bars are filled with red.

According to Figure 10, the error bars of the data are all filled with green. This proves that the difference between the observed water level and the calculated water level of the model is within the confidence interval and that the groundwater flow model is reliable. This means that the model can truly reflect the changes in the groundwater flow field in the study area and can be directly used in the ammonia migration model.

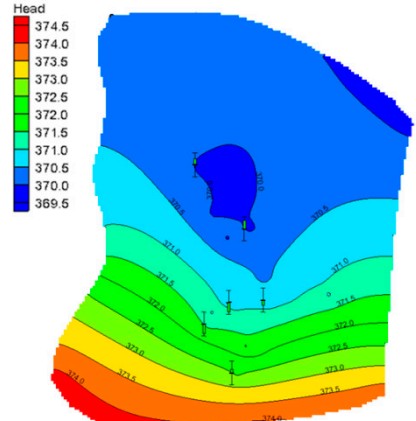

**Figure 10.** Fitting effect of the simulated water level and the measured water level.

### 2.3. Model of Migration of Pollutants in the Groundwater

On the basis of the above model, the model of migration of pollutants in the groundwater can be established by using MT3DMS of GMS. The partial differential equation of the model of the convection-diffusion of pollutants in saturated-unsaturated soils is as follows:

$$
\begin{cases}
\dfrac{\partial(\theta C)}{\partial u} = \dfrac{\partial}{\partial x_i}\left(\theta D_{ij}\dfrac{\partial C}{\partial x_i}\right) - \dfrac{\partial}{\partial x_i}(\theta v_i C) + q_s C_s + \sum R_n \\[2mm]
C(x,y,z,t) = c_0(x,y,z) \in \Omega, \qquad\qquad\qquad\qquad t = 0 \\[2mm]
C(x,y,z,t) = c(x,y,z) \in \Gamma_1, \qquad\qquad\qquad\quad t \geq 0
\end{cases}
\tag{6}
$$

where each part of the equation is as follows:

C—concentration of the dissolved phase of the pollutants ($ML^{-3}$);
$\theta$—porosity of the stratum medium (dimensionless);
t—time (T);
$x_i$—distance along the axis of the rectangular coordinate system (L);
$D_{ij}$—tensor of the coefficient of hydrodynamic dispersion ($L^2T^{-1}$);
$v_i$—actual flow rate of pore water on average ($LT^{-1}$);
$q_s$—flow of the aquifer per unit volume, representing source (positive) and sink (negative) ($T^{-1}$);

$C_s$—concentration of pollutants in a source or sink ($ML^{-3}$);
$\sum R_n$—term of chemical reaction ($ML^{-3}T^{-1}$); and
$\Omega$—overall simulation area.

### 2.3.1. Selection of Simulated Factors

According to the hydraulic gradient of the groundwater along the Feng River, four river profiles perpendicular to the flow direction of the groundwater were selected in the study area and divided the river into five sections. The four profiles are A, B, C, and D from south to north (Figure 11).

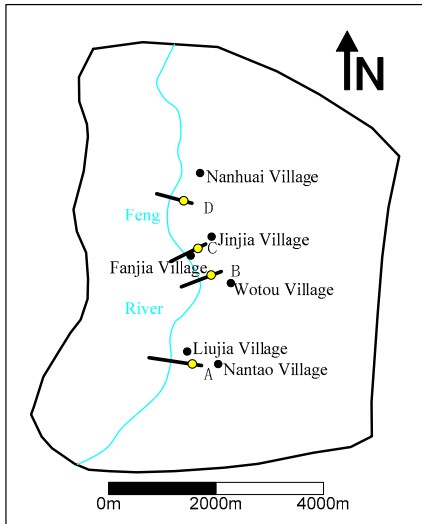

**Figure 11.** Positions of the four profiles.

Each profile was sampled, the concentration of ammonia in the river was measured, and the concentration was generalized as the input concentration at the upper boundary of the ammonia migration model (Table 3).

**Table 3.** Concentrations of ammonia in each profile.

| Profile Number | A | B | C | D |
|---|---|---|---|---|
| Concentration (mg/L) | 0.48 | 0.46 | 0.34 | 0.41 |

### 2.3.2. Stress Period

The simulation started from 1 January 2021 to 31 December 2040. Each corresponding calendar month was taken as a stress period, and then, the whole simulation period was divided into 240 stress periods.

## 3. Results

### 3.1. The Area of Influence of Ammonia

The area of influence of ammonia from the Feng River on the groundwater was simulated (Figure 12).

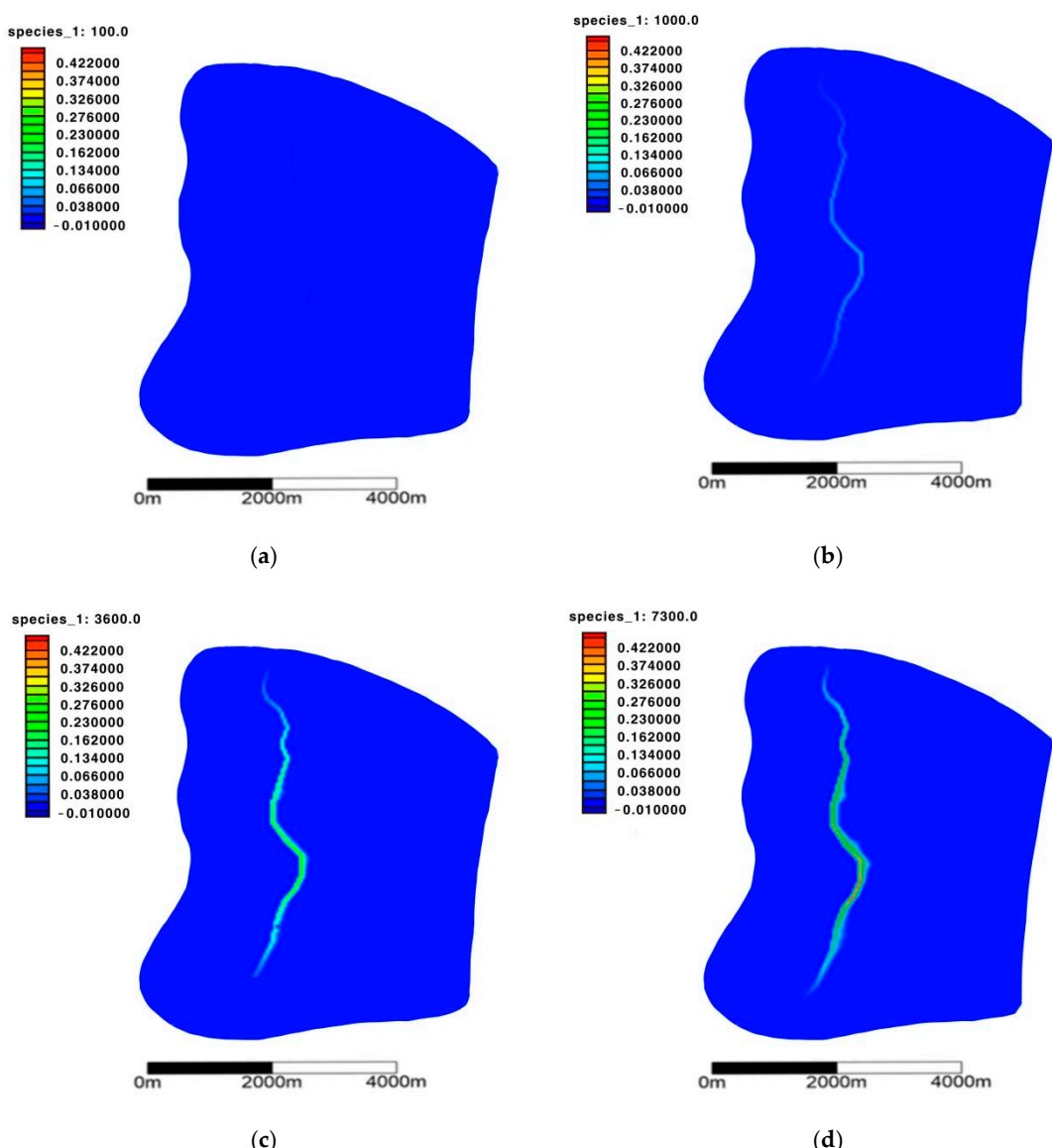

**Figure 12.** Simulation results of plane migration of the ammonia pollution plume. (**a**) Simulation results of plane migration of the ammonia pollution plume for 100 days (**b**) Simulation results of plane migration of the ammonia pollution plume for 1000 days (**c**) Simulation results of plane migration of the ammonia pollution plume for 3600 days (**d**) Simulation results of plane migration of the ammonia pollution plume for 7300 days.

### 3.2. The Ammonia Pollution Plume

The center of the ammonia pollution plume gradually moves eastward with the flow of groundwater over time. The migration of ammonia is similar to that of the center of ammonia pollution plume, and ammonia accumulates in the lower part of the river. During the simulation period, the ammonia pollution plume gradually becomes wider and the diffusion range of ammonia in the groundwater increases (Figure 13).

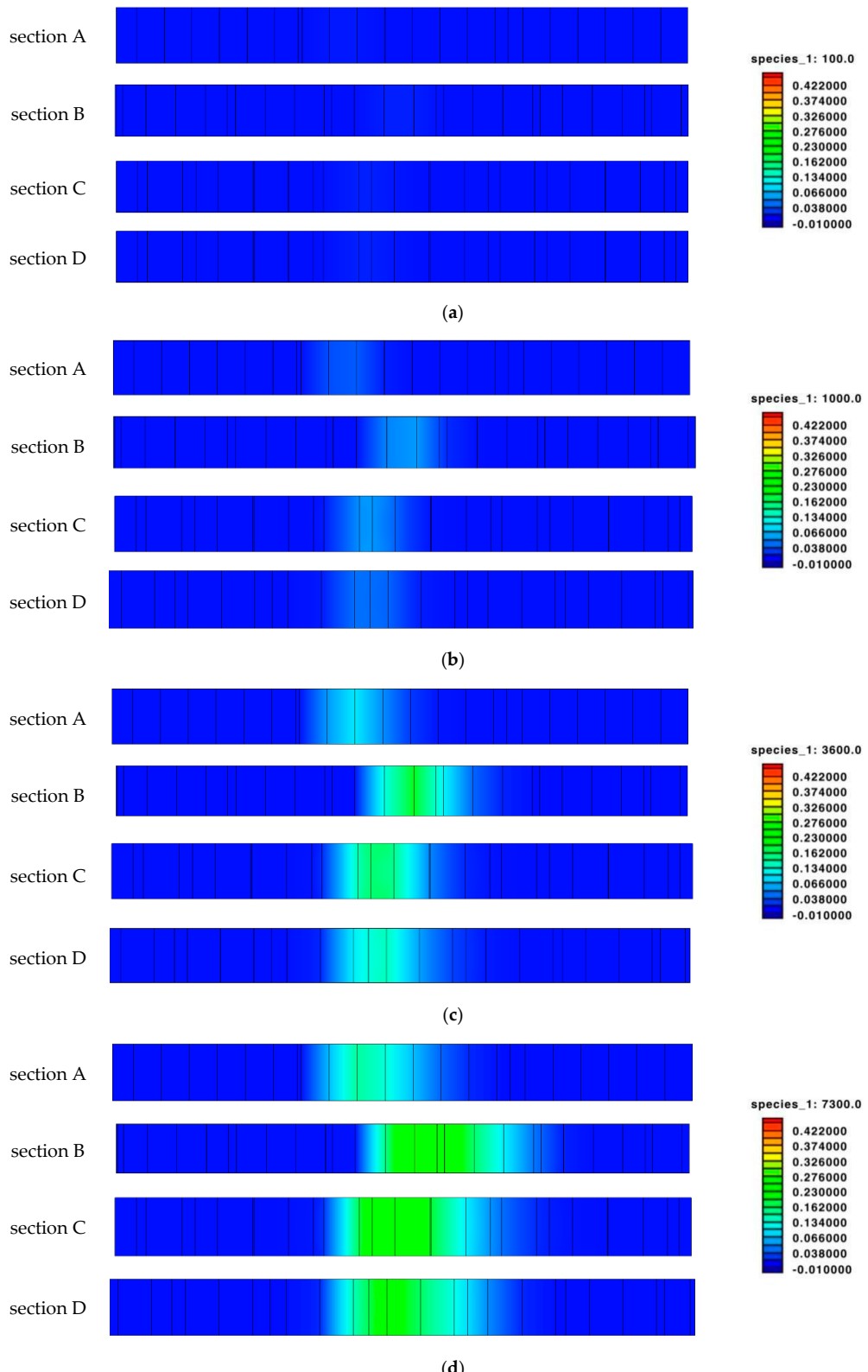

**Figure 13.** Simulation results of migration of the ammonia pollution plume in the four sections. (**a**) Simulation results of plane migration of ammonia pollution (**b**) Simulation results of plane migration of ammonia pollution (**c**) Simulation results of plane migration of ammonia pollution (**d**) Simulation results of plane migration of ammonia pollution.

### 3.3. The Ammonia Content

Well A at about 50 m from the river way and well B at about 200 m from the river way were taken as the objects of observation to observe the changes in ammonia content near wells during the simulation period (Figure 14).

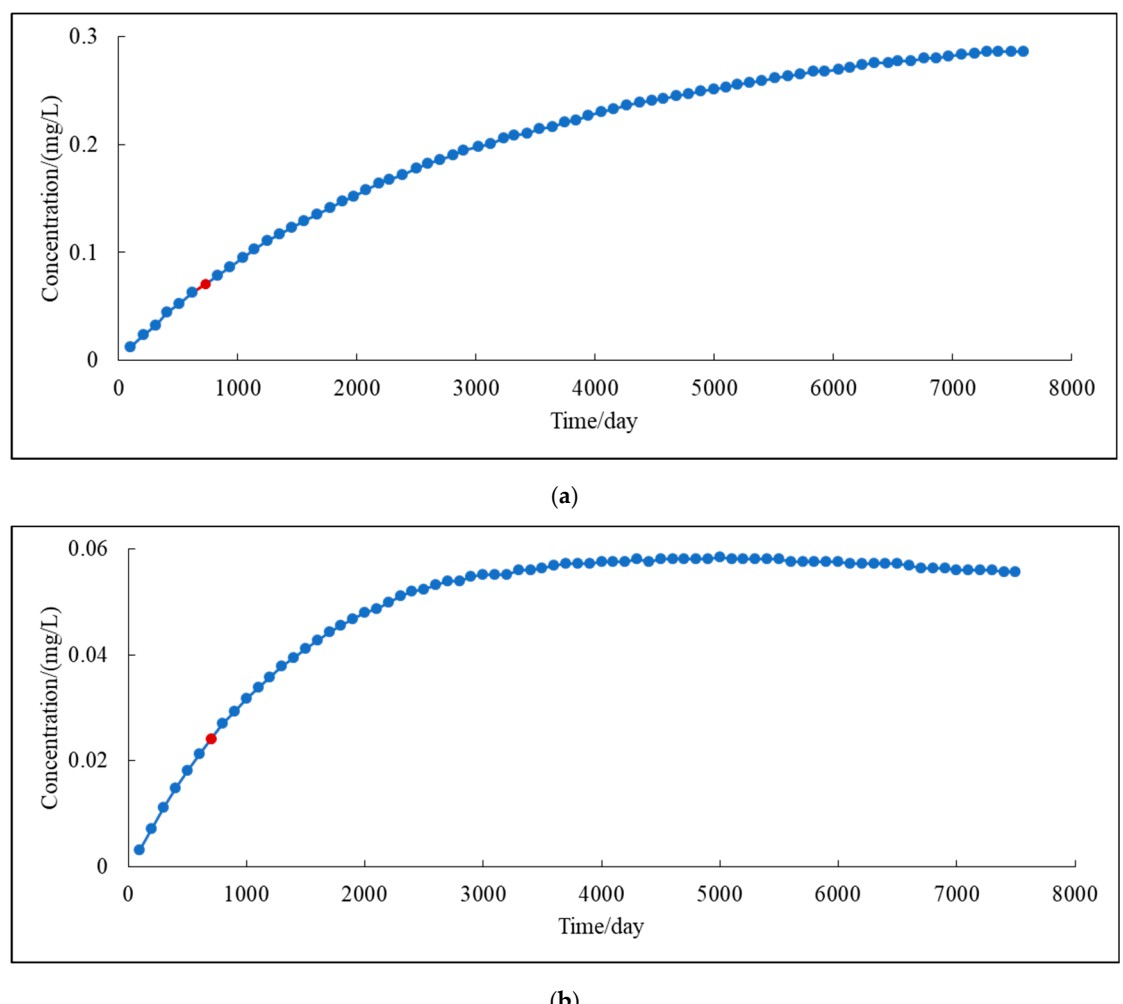

**Figure 14.** Changes in the ammonia content near wells. (**a**) Changes in ammonia content near well A. (**b**) Changes in ammonia content near well B.

The ammonia content near the wells close to the watercourse is greatly affected by the river. At the beginning of the simulation, ammonia in the river migrates to the wells. The ammonia content near well A stabilizes at about 0.30 mg/L around the year 2040, while the ammonia content near well B stabilized at about 0.06 mg/L around the year 2031. In general, the influence of ammonia from the river on the groundwater is within an acceptable range, and the ammonia content near the wells meets the Quality Standard for Ground Water III [ammonia (GB/T14848-2017 (in Chinese)].

## 4. Discussion

The reasons for the above situation are as follows:

In a water body, ammonia is an important parameter indicative of inorganic pollution that mainly result from anthropogenic impacts [34,35]. It was used as the primary restrictive descriptor for the water pollution control goals in the 12th Five-Year Plan of China (2011–2015) [36]. The reference ammonia concentration value established by the Quality Standard for Ground Water class III (GB/T14848-2017 (in Chinese)) is 0.5 mg/L. The sampling sites of this study are all located 5–12 m downstream of the village's sewage outfall in urban

areas and are presumably influenced by industrial effluent or domestic sewage discharge, which may contribute to the increased ammonia levels at these points. Apart from that, poultry, livestock, and agricultural fertilizers also attribute to the increase in ammonia.

In a town in Kunshan, China [37], the migration of groundwater pollution gradually spreads from the central area to the surrounding area, and the research center has the highest concentration and gradually decreases along the diffusion direction. This is because the surface elevation of the river valley is lower than that of other areas, and the groundwater level here is lower than that of other areas as well. Meanwhile, the contour lines under the river valley are fewer and the groundwater flows relatively slower. Therefore, as ammonia in the river migrates to groundwater under the river way, it accumulates. Different from another study, this study detected that, due to the pumping of groundwater at 50,000 t/d since 1958, when Xi'an the 3rd waterworks established the east Feng River bank, the groundwater level on the east bank of the Feng River dropped and a regional depression cone formed around the wells. Then, the ammonia in the river seeps into the groundwater and moves toward the east bank of the Feng River with the groundwater flow. Finally, it accumulates around the depression cone of the groundwater on the east bank of the Feng River, leading to an increase in the concentration of ammonia in the groundwater on the east bank of the Feng River, with the main impact range within 200 m along the Feng River.

Apart from domestic pollutants from residents along the river, waste water, and the industrial "three wastes" resulting in ammonia concentration in the river, this study finds that nitrogen fertilizers and sewage irrigation also have varying degrees of impact on the ammonia content of groundwater since the study area is located among farmland. Therefore, the ammonia content of the groundwater sampling points far from the watercourse is higher than that closer to the watercourse.

## 5. Conclusions

(1) During this simulation, a field survey was conducted in the study area. Combined with the current hydrology and water quality of the study area, the hydrogeological conceptual model was converted into a numerical model of groundwater flow and solute transport. Additionally, by using ammonia as the simulation factor, the calibrated model was used to simulate the hydrology and water quality of the study area. It is concluded that the established hydrogeological conceptual model and numerical model are correct, and the selected parameters and calculated source and sink terms are reasonable, which conform to the actual groundwater conditions in the study area. Thus, the results of this study can be used for water flow field research and water source mining planning.

(2) In this study, the GMS model was used to predict and analyze the diffusion process of pollutants in the study area and to simulate the migration trend for 20 years. Through the analysis of the GMS model, it is predicted that pollutants will gradually migrate eastward from the river to groundwater, with the impact of ammonia in the river water on groundwater remaining within an acceptable range, and the ammonia content at the intake well meets the Quality Standard for Ground Water class III [ammonia (GB/T14848-2017 (in Chinese))].

(3) Using the GMS model to simulate and predict the hydrological and water quality status of groundwater is conducive to understanding the status of groundwater pollution and to adopting thoughtful treatment and maintenance measures. However, the speed of pollutant migration and the size of the diffusion range are closely related to the amount and intensity of groundwater extraction. Therefore, when predicting the migration of pollutants, the influencing factors should be taken into consideration.

**Author Contributions:** Formal analysis, methodology, software, visualization, writing—original draft, writing—review and editing, Z.W.; funding acquisition, methodology, writing—review and editing, X.Z.; writing—review and editing, T.X.; data curation, N.W.; investigation, J.Y. All authors have read and agreed to the published version of the manuscript.

**Funding:** This research was funded by China postdoctoral science foundation, grant number 2018M643689.

**Institutional Review Board Statement:** Not applicable.

**Informed Consent Statement:** Not applicable.

**Data Availability Statement:** Part of the data (observations/measurements) used in this paper was gotten from field study or experiments conducted by my team. The other part of data was taken after application from the third waterworks in Xi'an and we are responsible for these data. The results of this study are freely available.

**Acknowledgments:** The third waterworks in Xi'an, China, provided the pumping wells and water quality data.

**Conflicts of Interest:** The authors declare no conflict of interest.

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
