# Peer review of "A Comprehensive Evaluation Model of Ammonia Pollution Trends in a Groundwater Source Area along a River in Residential Areas"

_water, doi:10.3390/w13141924_

Round 1

Reviewer 1 Report

This paper presents a case study of groundwater quality evaluation in the Feng (or Fenghe??, because on line 82 is written “Feng” and on line 119 is written Fenghe,the authors must write the name of the river in the same way) River area (residential areas) in China: ammonia nitrogen pollution trend analysis. The methods applied are “coupling models and their interrelated models, including (i) MODFLOW and (ii) MT3DMS”.

The paper presents a work with public utility, in the area of groundwater quality management. However, the acceptance of the manuscript for publication would depend on major revisions. The author needs to provide a point by point response or provide a rebuttal if some of the reviewer’s comments cannot be revised.

Here is my justification for the recommendation:

Sometimes there is lack of rigor and the authors present different processes (models and numerical simulations), but with little justification. They must better justify their chosen processes and namely “coupling models and their interrelated models”, explaining better “their interrelated models”.

Also, this paper needs a major revision of language, it is full of grammatical errors, especially regarding subject-verb agreement. Some examples:

1-Title: The comprehensive evaluation model of ammonia nitrogen pollution trend in groundwater source area along river in residential areas = The comprehensive evaluation model of ammonia nitrogen pollution trend in groundwater source area along a river in residential areas

2-Abstract: in a plain along river where = in a plain along a river, where

That means the model =This means that the model

which is less than 200 meters from river= which is less than 200 meters from the river

3-Introduction

fertilizers[6] = fertilizers [6] (this kind of references without a space before the brackets is constant througouht the paper)

which consumes oxygen in water and destroy the balance = it consumes oxygen in water and destroys the balance

can stimulate human eyes = can stimulate affect human eyes

main sources of water for industrial, agricultural and life = main sources of water for industrial, agricultural and life purposes

The authors should inform the reader about which software or softwares they used in this study. I think it was the Aquaveo's software, because in the site of the Aquaveo's software they say “Our name is synonymous with visualizing water. We provide solutions for modeling and visualizing groundwater & surface-water hydrology and hydraulics.” And the authors said in the Abstract “It consists of coupling models and their interrelated models, including (i) MODFLOW and (ii) MT3DMS”. The authors should inform if they used these models of Groundwater Modeling System (GMS) of this software company, and, in particular, explain better which softwares were used (structured or unstructured grid points?). For instance, clarifying if MODFLOW is a cell-centered saturated flow model that can perform both steady state and transient analysis and has a wide variety of boundary conditions and input options. MT3DMS is a modular three-dimensional transport model for the simulation of advection, dispersion, and chemical reactions of dissolved constituents in groundwater. Both MODFLOW and MT3DMS are available in Aquaveo's software.

Section 2.2 Numerical Simulation of Groundwater Flow: this section with their subsections are not clear, because the reader can not make the connection between them and the justifications of the classifications are missing (why section A, B, C, and D? It was based on Table 3, but why did you use those concentration (mg/L) values for the classification?). The authors calculate Qrain, Qriver, QLateral, and Qirrigation in this section. For example, what is the purpose of Q values? What were they calculated/simulaled for? For example, by using the Qrain, you obtain the partition I and II, and what criteria did you used to consider those parameter values to classify the partition? How did you obtain the Table 3 values? The paper is confused, and needs more justifications.

Figures 3, 4, 5, 8, 13 and 14 are not referenced throughout the text.

Figures 1, 2, 8, 11 and 14 are not legible, in particular the "Legends" of these figures cannot be read. The authors should increase their font size.

On line 54 where is “figure 1” should be “Figure 1”.

On line 116 where is “GLM” should be “Groundwater Modeling System (GMS)”.

Author Response

Dear Sir/Madam,

Thanks for your comments and they are very important to my paper writing and my work. We are very sorry for the ambiguity that our unintelligible description brings for you to better understand our research. We have double checked the manuscript and corrected some stylistic and orthographic errors in this paper with the help of an English philologist, which we hope meet with the reviewer’s approval. Following responses are the replies to your comments:

Point 1: This paper presents a case study of groundwater quality evaluation in the Feng (or Fenghe??, because on line 82 is written “Feng” and on line 119 is written Fenghe,the authors must write the name of the river in the same way) River area (residential areas) in China: ammonia nitrogen pollution trend analysis. The methods applied are “coupling models and their interrelated models, including (i) MODFLOW and (ii) MT3DMS”.

Response 1: Line 270 has been corrected in the newly submitted manuscript in which Fenghe River has been changed into Feng River using the “Track Changes” function.

Point 2: Sometimes there is lack of rigor and the authors present different processes (models and numerical simulations), but with little justification. They must better justify their chosen processes and namely “coupling models and their interrelated models”, explaining better “their interrelated models”.

Response 2: GMS is a 3D groundwater modeling system software which is developed by Institute of environmental simulation, University of Birmingham, USA and US Army drainage engineering test station. MT3DMS of GMS is a 3D solute transfer model which simulates convection-diffusion and chemical reactions in groundwater system by using the MODFLOW together. Considering the Ammonia in surface water would transfer to the upper boundary of the groundwater causing pollution, thus this paper adopts MODFLOW and MT3DMS. [TAN Wenqing, SUN Chun, HU Jingmin, et al. The application of GMS in the numerical simulation prediction of groundwater pollution [J]. Water Resources & Hydropower of Northeast China, 2008, (5). (In Chinese)]

Point 3: Also, this paper needs a major revision of language; it is full of grammatical errors, especially regarding subject-verb agreement. Some examples:

1-Title: The comprehensive evaluation model of ammonia nitrogen pollution trend in groundwater source area along river in residential areas = The comprehensive evaluation model of ammonia nitrogen pollution trend in groundwater source area along a river in residential areas

2-Abstract: in a plain along river where = in a plain along a river, where

That means the model =This means that the model

which is less than 200 meters from river= which is less than 200 meters from the river

3-Introduction

fertilizers[6] = fertilizers [6] (this kind of references without a space before the brackets is constant throughout the paper)

which consumes oxygen in water and destroy the balance = it consumes oxygen in water and destroys the balance

can stimulate human eyes = can stimulate affect human eyes

main sources of water for industrial, agricultural and life = main sources of water for industrial, agricultural and life purposes

Response 3: The language problems you mentioned have been revised word by word in the newly submitted manuscript using the “Track Changes” function with the help of and English philologist.

Point 4: The authors should inform the reader about which software or softwares they used in this study. I think it was the Aquaveo's software, because in the site of the Aquaveo's software they say “Our name is synonymous with visualizing water. We provide solutions for modeling and visualizing groundwater & surface-water hydrology and hydraulics.” And the authors said in the Abstract “It consists of coupling models and their interrelated models, including (i) MODFLOW and (ii) MT3DMS”. The authors should inform if they used these models of Groundwater Modeling System (GMS) of this software company, and, in particular, explain better which softwares were used (structured or unstructured grid points?). For instance, clarifying if MODFLOW is a cell-centered saturated flow model that can perform both steady state and transient analysis and has a wide variety of boundary conditions and input options. MT3DMS is a modular three-dimensional transport model for the simulation of advection, dispersion, and chemical reactions of dissolved constituents in groundwater. Both MODFLOW and MT3DMS are available in Aquaveo's software.

Response 4: Our research uses the GMS of Aquaveo’s software which is one of the most popular groundwater modeling software with clear interface, powerful pre-treatment and post-treatment functions and better 3D visualization effect. The usage of the GMS of Aquaveo's software has been mentioned in lines 90-97 in the newly submitted version. The study adopted structured grid points because boundary fitting could be easily achieved and it applies to the calculation of fluid and surface stress concentration, etc. Besides the structured grid points could generate grid points with high speed and better quality. 

Point 5: Section 2.2 Numerical Simulation of Groundwater Flow: this section with their subsections are not clear, because the reader can not make the connection between them and the justifications of the classifications are missing (why section A, B, C, and D? It was based on Table 3, but why did you use those concentration (mg/L) values for the classification?). The authors calculate Qrain, Qriver, QLateral, and Qirrigation in this section. For example, what is the purpose of Q values? What were they calculated/simulaled for? For example, by using the Qrain, you obtain the partition I and II, and what criteria did you used to consider those parameter values to classify the partition? How did you obtain the Table 3 values? The paper is confused, and needs more justifications.

Response 5: Our study divided the 4 sections first, and measured the concentration afterward. According to groundwater hydraulic gradient along the Feng River, we selected 4 sections perpendicular to the direction of groundwater flow in study area. The hydraulic gradients in these four sections are almost the same and they are all located 5-12m downstream of the village's sewage outfall.

The concentrations of Ammonia of the Feng River in sections A, B, C and D, are used as input ammonia concentration of upper boundary in transfer model. The concentrations in Table 3 are measured values.

The purpose of calculation Q values is to make a comparison between calculated values and measured values to check if there are any deviations.  If the deviations are within the allowable range, the simulation could continue.

Point 6: Figures 3, 4, 5, 8, 13 and 14 are not referenced throughout the text.

Response 6: This part had been revised in the newly submitted manuscript.

Point 7: Figures 1, 2, 8, 11 and 14 are not legible, in particular the "Legends" of these figures cannot be read. The authors should increase their font size.

Response 7: This part had been revised in the newly submitted manuscript.

Point 8: On line 54 where is “figure 1” should be “Figure 1”.

Response 8: This part had been revised in the newly submitted manuscript.

Point 9: On line 116 where is “GLM” should be “Groundwater Modeling System (GMS)”.

Response 9: This part had been revised in the newly submitted manuscript.

Thanks for your advice again and hope to learn more from you.

Reviewer 2 Report

Please see the attached review file.

Author Response

Dear Sir/Madam,

Thanks for your comments and they are very important to my paper writing and my work. We are very sorry for the ambiguity that our unintelligible description brings for you to better understand our research. We have double checked the manuscript and corrected some stylistic and orthographic errors in this paper with the help of an English philologist, which we hope meet with the reviewer’s approval. Following responses are the replies to your comments:

Point 1: Lines 28-38:

Change the formatting to normal text (lines 28-59).

This section is worded a bit too vaguely and generally. It should be more concrete. Also the style should be refined. Please specify, what kind of nitrogen-containing pollutants do you mean, add few numeriacal data.

Response 1: The formatting in lines 28-59 has been changed into normal text in the newly submitted manuscript.

Point 2: Lines 42-46: Please show, where these studies were completed.

Response 2: Lines 42-46, the reference 17 was conducted in Aberdeenshire Scotland. The reference 18 was conducted in Japan. The reference 18 was conducted in Plynlimon in the mid-Wales.

Point 3: Lines 48-50: When river water replenishes groundwater, solute in water enters groundwater or coastal pumping wells through river infiltration system under the action of natural existence

It is unclear that the authors mean. Please rewrite this sentence.

Response 3: Lines 57-58: “When river water replenishes groundwater, solute in river water enters groundwater or coastal pumping wells through river infiltration system by nature.” in newly submitted manuscript.

Point 4: Figure 1: The image is blurry.

Response 4: This part had been revised in the newly submitted manuscript.

Point 5: Lines 76-79: The novelty of this particular study has been insufficiently substanciated by the authors.

Response 5: Lines 91-98: “Thus, taking Feng River water source as the research object, this study simulates and analyses the hydrological and water quality characteristics and the trends of groundwater ammonia pollutants migration in the study area using MODFLOW and MT3DMS in GMS.  By field research and consulting the documents and of hydrogeological condition in the area, mathematical model of hydrology and water quality is established which is used in GMS to correct and validate main parameters. This particular study is beneficial to groundwater protection and provides basis for the scientific development and utilization.” in newly submitted manuscript.

Point 6: Figure 2: In addition to the study area map provided by the authors, a map showing the study area within China would be helpful.

Response 6:  This part had been revised in the newly submitted manuscript.

Point 7: Line 95: The vertical movement of the water flow is not considered...

On what basis this simplification was done?

Response 7: In the study area, the groundwater flow is mainly laminar flow. The vertical flow occupies less than 10% of laminar flow because the lower boundary water barrier is silty clay. According to Aviriyanover, the maximum buried depth of evaporation from the phreatic aquifer varies and ranges from 1.5 m to 4.0 m. As the buried depths of the phreatic aquifer in the study area are all over 4m, the evaporation capacity is viewed as zero, thus the vertical flow can be ignored.

Point 8: Table 1 How the parameter values were selected?

Response 8: Parameters were selected according to the depth of groundwater level and unsaturated zone lithology.

Point 9: Figure 8.Improve the visibility of text on the figure.

Response 9: This part had been revised in the newly submitted manuscript.

Point 10: Lines: 188-190 That would be 11 stress periods?

Response 10: It’s an input mistake. This part had been revised in the newly submitted manuscript.

Point 11: Lines 192-194: How do you take the climate change into consideration? Do you expect the precipitation increase or decrease in the study area and its vicinity? How do you expect the water use to change in this period?

Response 11: We haven’t take climate change into consideration in the study, but the impact the climate change brings should be studied in the further research. Thanks for your advice again, and i hope to learn more from your comments.

We hold that the precipitation will increase year by year for the northwest China is becoming warm and moist gradually which had been considered as the arid, semi-arid area. This viewpoint could be verified in the following 2 papers. (1)In the past 40 years, the temperature increase rate in the upper reaches of the Yellow River was 0.023℃/a, and the precipitation increase rate was 1.09㎜/a. (Ye P, Zhang Q, Wang Y, et al. Characteristics of climate change in the upper Yellow River Basin and its influence on vegetation and runoff during recent 40 Years [J]. Transactions of Atmospheric Sciences, 2020, (11). (In Chinese)). (2)The correlation analysis of the rainfall characteristics in Xi’an in the past 32 years shows that the annual precipitation in Xi’an has increased at a rate of 0.474 mm/a. (Li J, Wei Y. Analysis of Precipitation Change Characteristics in Xi'an [J]. Haihe Water Resources, 2019, (6). (In Chinese)).

With the development of economy and the expansion of the city, the study area is under the urbanization construction and transformation. A large number of people will flood into the new urban area, and the domestic water use will increase rapidly. However, agricultural production will disappear gradually in this area, the agricultural water use will plummet.

Point 12: Lines 253-255: These calculations are based on the current ammonia content in the river water, right? But how it has changed over time and how do you expect it to change in future?

Response 12: Yes, these calculations are based on the current ammonia content in the river water assuming the ammonia content is stable and infiltrate into ground water with stable ammonia content. Talking about the ammonia content in the future, as I have answered previously, when the agricultural production ceased with the urbanization in the future, the nitrogen fertilizer usage and nitrogen emissions from aquaculture will fall to 0.  And this process will only take 3-5 years.

Thanks for your advice again and hope to learn more from you. 

Reviewer 3 Report

Summary

The manuscript uses a novel model to predict the nutrient trends in groundwater. The data reveal that the model performs well and closely matches empirical data collected in National Pollution Source Census Reporting China. The data analyses in this manuscript  are very comprehensive.  A significant contribution of this paper is that it adds more to the body of literature that highlights the need for models to evaluate issues of water quality;  this is a very positive argument for publishing this paper. That said, I think this manuscript would require a complete overhaul before it is ready for publication.

Major comments

       I have three basic criticisms of the paper. First, I found that one of the aims of the study were hardly explained. Specifically,  why are models important and how do these findings relate to global issues surrounding water quality.  I think a rewriting of the introduction will be helpful. The second criticism regards grammar and layout of the, manuscript (especially the results, which is probably too long by a significant amount. As an author, I hate to get such criticism, but in this case, I believe that the main message gets lost in the detail of the results section. This is something I often face with my manuscripts, but in the end, trimming usually produces a concise paper with the essential details that the readers are likely to absorb. Third, the discussion is really weak and  hardly explains you results relative to other models used to trace nutrient concentrations.

Minor comments

So many grammatical issues I could not write all of them here.

The comprehensive evaluation model of ammonia nitrogen pollution trend in groundwater source area along river in resi-dential areas

Lines 1- 2: The title above does is grammatical incorrect. Change your title to reflect your major findings

 Line 11: change trend to trends

Line 12: was should be is

Ammonia nitrogen? I do not what these means

You graphs need better resolution; I can hardly see the simulation properly

Author Response

Dear Sir/Madam,

Thanks for your comments and they are very important to my paper writing and my work. We are very sorry for the ambiguity that our unintelligible description brings for you to better understand our research. We have double checked the manuscript and corrected some stylistic and orthographic errors in this paper with the help of an English philologist, which we hope meet with the reviewer’s approval. Following responses are the replies to your comments:

Point 1: First, I found that one of the aims of the study were hardly explained. Specifically, why are models important and how do these findings relate to global issues surrounding water quality. I think a rewriting of the introduction will be helpful.

Response 1: We have further explained the aim of our study and revised the introduction part in the newly submitted manuscript.

Point 2: The second criticism regards grammar and layout of the, manuscript (especially the results, which is probably too long by a significant amount. As an author, I hate to get such criticism, but in this case, I believe that the main message gets lost in the detail of the results section. This is something I often face with my manuscripts, but in the end, trimming usually produces a concise paper with the essential details that the readers are likely to absorb.

Response 2: The workload of this research is relatively large, and the calculation results are relatively excessive. And these data and results play an essential role on my final conclusions. Therefore, this paper concluding all the research process to make it logical and reasonable is too long for readers. Thank you for your suggestions, we will trim the text in future paper writing so that readers can understand the core content easily.

Point 3: Third, the discussion is really weak and hardly explains you results relative to other models used to trace nutrient concentrations.

Response 3: We have added some discussion in this part.

Point 4: Lines 1- 2: The title above does is grammatical incorrect. Change your title to reflect your major findings.

Response 4: This part had been revised in the newly submitted manuscript.

Point 5: Line 11: change trend to trends

Response 5: This part had been revised in the newly submitted manuscript.

Point 6: Line 12: was should be is

Response 6: This part had been revised in the newly submitted manuscript.

Point 7: Ammonia nitrogen? I do not what these means

Response 7: Ammonia equals to Ammonia nitrogen (NH4+-N). Some papers use ammonia and others use ammonia nitrogen.

Point 8: You graphs need better resolution; I can hardly see the simulation properly

Response 8: This part had been revised in the newly submitted manuscript.

Thanks for your advice again and hope to learn more from you.

Round 2

Reviewer 2 Report

The authors have addressed the issues raised by the reviewer and the revised manuscript shows improvement in comparison with the original submission. Therefore I agree with the acceptance of this manuscript for publication.

Author Response

Thanks again for you constructive suggestions.

Reviewer 3 Report

I want to applaud the authors for addressing the comments adequately. Some issues are minor grammatical errors.

I still think the discussion needs a little more detail - especially about the broader implications of the authors' findings

Author Response

Thanks for your comments and I learnt a lot from your suggestions.